# Peptosome Coadministration Improves Nanoparticle Delivery to Tumors through NRP1-Mediated Co-Endocytosis

**DOI:** 10.3390/biom9050172

**Published:** 2019-05-05

**Authors:** Zhichu Xiang, Gexuan Jiang, Xiaoliang Yang, Di Fan, Xiaohui Nan, Dan Li, Zhiyuan Hu, Qiaojun Fang

**Affiliations:** 1CAS Key Laboratory for Biomedical Effects of Nanomaterials & Nanosafety, National Center for Nanoscience and Technology, Beijing 100190, China; xiangzc@nanoctr.cn (Z.X.); jianggx@nanoctr.cn (G.J.); xiaoliang.yang1@gmail.com (X.Y.); fand2018@nanoctr.cn (D.F.); nanxh@nanoctr.cn (X.N.); lid@nanoctr.cn (D.L.); 2CAS Center for Excellence in Nanoscience, National Center for Nanoscience and Technology, Beijing 100190, China; 3University of Chinese Academy of Sciences, Beijing 100049, China; 4Sino-Danish Center for Education and Research, Beijing 101408, China; 5Beijing Key Laboratory of Ambient Particles Health Effects and Prevention Techniques, National Center for Nanoscience and Technology, Beijing 100190, China

**Keywords:** peptosome, self-assembly, nanoparticle delivery, coadministration, co-endocytosis

## Abstract

Improving the efficacy of nanoparticles (NPs) delivery to tumors is critical for cancer diagnosis and therapy. In our previous work, amphiphilic peptide APPA self-assembled nanocarriers were designed and constructed for cargo delivery to tumors with high efficiency. In this study, we explore the use of APPA self-assembled peptosomes as a nanoparticle adjuvant to enhance the delivery of nanoparticles and antibodies to integrin αvβ3 and neuropilin-1 (NRP1) positive tumors. The enhanced tumor delivery of coadministered NPs was confirmed by better magnetosome (Mag)-based T_2_-weighted magnetic resonance imaging (MRI), liposome-based fluorescence imaging, as well as the improved anti-tumor efficacy of monoclonal antibodies (trastuzumab in this case) and doxorubicin (DOX)-containing liposomes. Interestingly, the improvement is most significant for the delivering of compounds that have active or passive tumor targeting ability, such as antibodies or NPs that have enhanced permeability and retention (EPR) effect. However, for non-targeting small molecules, the effect is not significant. In vitro and in vivo studies suggest that both peptosomes and the coadministered compounds might be internalized into cells through a NRP1 mediated co-endocytosis (CoE) pathway. The improved delivery of coadministered NPs and antibodies to tumors suggests that the coadministration with APPA self-assembled peptosomes could be a valuable approach for advancing αvβ3 and NRP1 positive tumors diagnosis and therapy.

## 1. Introduction

Nanocarriers loaded with drugs and probes are expected to improve tumor diagnosis and treatment while reducing the side effects. However, the applications are strongly impeded by their low delivery efficacy to tumors [1,2]. In addition to vascular walls and cell membranes, heterogeneous and dense tumor stroma also hinders the permeability of nanoparticles (NPs) into tumors [3,4,5]. The penetration efficacy can be improved by charge modification to increase electrostatic interactions or functionalizing the surface of NPs with cell-penetrating peptides (e.g., TAT or iRGD peptides) [6,7]. Unfortunately, the electrostatic attraction leads to nonspecific cellular uptake, and the modification with cell-penetrating peptides on the surface of NPs can be complicated [8].

Nanoparticle carriers, especially biodegradable nanocarriers, have been widely studied in the field of anticancer drug delivery due to their high drug-loading capacity, biocompatibility, and stability [9,10]. To improve tumor specificity and drug-loading capacity, nanocarriers are usually complicatedly designed which limits their application in clinical practice [11]. An alternative solution is using a simple compound as an adjuvant to assist the nanocarriers delivering specifically to tumors. One example of such a compound is the iRGD (CRGDKGPDC) peptide [12,13], which contains a consensus R/KXXR/K motif (CendR motif) and becomes active after being proteolytically cleaved at the tumor site [14]. The receptor of the R/KXXR/K motif is neuropilin-1 (NRP1), a cell surface receptor that plays a key role in the angiogenesis and regulation of vascular permeability [15,16,17,18]. The iRGD peptide has been widely used to modify NPs or coadministrate with other reagents for effective tumor targeting and penetration [19,20,21,22]. Unfortunately, the plasma half-life of iRGD is rather short which may impede its application as an adjuvant [23].

In our previous study, a tumor-targeting and penetrating nanosystem based on the amphiphilic peptide PA (APPA: Ac-RGDDK(C18)CK(C18)DRGDK-COOH) that contains both RGD and CendR motifs was designed and constructed (https://onlinelibrary.wiley.com/doi/10.1002/adtp.201800107). The nanosystem homes to tumor vessels through the binding of RGD with integrin αvβ3 and then penetrates into tumor tissue through the interaction of the CendR motif with NRP1 receptor [12]. Since both αvβ3 and NRP1 are highly expressed in many tumors [24] the nanosystem could have wide applications in cancer therapy. In this study, we further demonstrate that the APPA self-assembled peptosomes can be used as an adjuvant to improve the delivery of antibodies, nanoprobes, and nanodrugs to tumors. We suggest that such a coadministration strategy can be a straightforward and effective approach for clinical cancer diagnosis and therapy.

## 2. Materials and Methods

### 2.1. Materials

The octadecanoic acid chain (C18) modified amphiphilic peptide PA, peptide iRGD, and control peptide PC were synthesized by Top-peptide Bio Co., Ltd. (Shanghai, China) using a Fmoc strategy solid phase peptide synthesis (SPPS) method. Magnetotactic bacteria, *M. gryphiswaldense* (MSR-1), were provided by China Agricultural University. Fluorescein isothiocyanate (FITC), Hoechst 33342, (3-(4,5-dimethylthiazol-2-yl)-2,5-diphenyltetrazolium bromide (MTT), Tetramethylrhodamine isothiocyanate mixed isomers (TRITC), Poly(d,l-lactide-co-glycolide) (PLGA, MW 7–17K), lecithin, and dimethyl sulfoxide (DMSO) were purchased from Sigma Aldrich (St. Louis, MO, USA). Phosphate-buffered saline (PBS), Dulbecco’s modified eagle medium (DMEM), and RPMI 1640 medium were purchased from Hyclone. Fetal bovine serum (FBS) was purchased from Gibco. Doxorubicin (DOX) was purchased from KeyGEN Biotech (Jiangsu, China). DiR (DilC18(7)) and Cy5 were purchased from Life Technologies (Carlsbad, CA, USA). Trastuzumab was purchased from Genentech (Sanfrancisco, CA, USA). All antibodies used in this study were purchased from Cell Signaling Technology (Danvers, MA, USA). 1,2-distearoyl-sn-glycero-3-phospho -ethanolamine-*N*-[methoxy(polyethylene glycol)-2000] (DSPE-PEG) was purchased from Avanti (Alabaster, AL, USA). Fe_3_O_4_ nanoparticles were purchased from Nanoeast Biotech (Nanjing, China). The 4T1, HUVEC, and Hela cell lines were bought from ATCC (ATCC CRL-2539, ATCC CRL-1730, ATCC CCL-2). PC-3 and BT474 cell lines were bought from NICLR (National Infrastructure of Cell Line Resource, China) (No.3111C0001CCC000115, No.3142C0001000000314). The mice used in this study were bought from Charles River (Beijing, China).

### 2.2. Self-Assembly of Amphiphilic Peptide and Peptosome Construction

For the construction of APPA and APPC self-assembled peptosomes, 1 mg of all three compounds that were to be encapsulated in peptosomes (DOX, DiR, or TRITC) were weighed by a Precision Electronic Balance (Sartorius-BT25S). Each compound was dissolved in 100 uL of DMSO to make a stock solution at 10mg/mL. Then 1 mg of peptide was dissolved in the above stock solution in the amount of 5 μL of DOX, 2 μL of DiR, and 10 μL of TRITC, respectively. DMSO was added to bring up the solution to 10 uL, and 1 mL of PBS was added to the mixture and ultrasonicated (50 W) for 10 min, followed by incubation at room temperature for 1 h. The final solution was centrifuged at 5500× *g* for 5 min and the aqueous solution was collected. Transmission electron microscopy (TEM, HT7700, Hitachi, Japan) was used to characterize the morphology and the size of APPA and APPC self-assembled peptosomes [24]. Briefly, the collected aqueous solution was dropped onto a carbon grid, dried, and negatively stained with uranyl acetate for TEM. Dynamic light scattering (DLS, Zetasizer Nano ZS90, Malvern) was used to check the size distribution and surface zeta potential of the self-assembled nanoparticles.

### 2.3. Preparation of Magnetotactic Bacteria and Magnetosomes

The magnetotactic bacteria, MSR-1, were grown in a shaking flask at 30 °C and 100 r/m with the culture medium as described in our previous study [25]. The bacteria were harvested by centrifugation at 8000× *g* for 15 min and the supernatant was discarded. To extract magnetosomes, 1 mL of cell pellet was suspended in 2 mL PBS (10 mM, PH 7.4) and sonicated for 26 min at 3 s per time with an interval of 5 s at 200 W to lyze the cells. Afterwards, the suspension of magnetosomes was magnetically separated using a strong magnet, then the supernatant was removed and the separated magnetosomes were resuspended in 3 mL PBS (10 mM, PH 7.4) and sonicated for 13 min at 3 s per time with the time interval of 5 s at 120 W. The suspension was placed on the magnet to separate the magnetosomes. This procedure was repeated twice with the power changing to 80 W and 40 W, respectively. The magnetosomes were washed 3 times with PBS (10 mM, PH 7.4) and characterized with TEM. Dynamic light scattering was used to check the size and surface zeta potential of magnetosome nanoparticles.

### 2.4. Preparation of iRGD-Cy5, LipoDOX, and LipoCy5

The Cy5-NHS was used to label the iRGD peptide by the reaction of NHS with the amino group of iRGD. The reaction was conducted by mixing 0.3 mg of Cy5-NHS and 0.5 mg of iRGD (molar ratio 1:1) in 1 mL sodium bicarbonate buffer (pH 6.5). Then the Cy5 labeled peptide were purified by using a Hitachi HPLC system (L-7100, Hitachi, Japan). The Doxorubicin (DOX) and Cy5 loaded liposomes were prepared based on a modified single step preparation method [26]. Stock solutions of DSPE-PEG and lecithin were prepared separately at the concentrations of 1 mg/mL in 4% ethanol aqueous solution. Stock solutions of PLGA were prepared at the concentration of 2.5 g/mL in acetone. DOX and Cy5 were dissolved in water at the concentrations of 1 mg/mL and 0.1 mg/mL, respectively. The stock solutions of DSPE-PEG and lecithin were added into ddH_2_O and then the PLGA solution mixed with DOX or Cy5 was carefully pipetted into the resulting aqueous solution under sonication for 20 min at 100 W. The ratio of the used amount of aqueous solution to that of organic solution was 10:1. The prepared nanoparticles were purified and washed three times with a centrifugal filter (MW: 10 K) and characterized with TEM. Dynamic light scattering was used to analyze the size and surface zeta potential of the prepared nanoparticles.

### 2.5. Plasma Half-Life Comparison of Peptosome PADiR and Peptide iRGD

All animal experiments were conducted in accordance with the institutional guidelines approved by the Institutional Ethical Committee of Animal Experimentation of National Center for Nanoscience and Technology. Animals received care in accordance with the Guidance Suggestions for the Care and Use of Laboratory Animals (SCXK2016-0048). To compare the plasma half-life of peptosome PADiR and peptide iRGD, free Cy5, PADiR, and iRGD (Cy5 labeled) were intravenously injected into three 5–7 week-old female BALB/c nude mice, respectively. A total of 30 μL of blood from each mouse was collected 1 h, 2 h, 3 h, 5 h, 8 h, 10 h, 12 h, and 24 h after the injection and the fluorescence signal was detected using an IVIS SPECTRUM in vivo imaging system (PerkinElmer, Waltham, MA, USA). Cy5 was excited by a 640-nm laser and collected between 650 and 680 nm while DiR had excitation at 745 nm and emission at 800 nm.

### 2.6. Cell Culture and Tumor Model Construction

Five different cell lines were used in this study. 4T1, HUVEC, PC-3, and BT474 human breast cancer cells all have a high expression of both integrin αvβ3 and neuropilin-1 (NRP1) receptor. HeLa cells that express integrin αvβ3 but not NRP1 were used as the negative control [6,27]. 4T1 cells were cultured in RPMI 1640 medium (Hyclone) supplemented with 10% fetal bovine serum (Gibco). The HUVEC, PC-3, and HeLa cells were cultured in DMEM/High glucose (Hyclone) medium supplemented with 10% fetal bovine serum (Gibco). The BT474 cells were cultured in SFM4MAB medium supplemented with 10% fetal bovine serum (Gibco).

For the construction of 4T1 orthotopic tumors, 1 × 10^6^ cells were orthotopically injected into the 5–7 week-old female BALB/c mice. For the construction of PC-3 and HeLa xenograft tumors, 1 × 10^7^ cells were injected into the right/left flank of the 5–7 week-old female BALB/c nude mice. When the tumors were approximately 150 mm^3^ in size, the in vivo fluorescence imaging experiments were conducted.

### 2.7. Western Blot Analysis

The cells were collected and lysed with RIPA buffer (Solarbio, Beijing, China) containing 1 mM phenylmethanesulfonyl fluoride (PMSF) (Solarbio, Beijing, China). Protein samples (50–80 μg) were electrophoresed on 6% sodium dodecyl sulfate–polyacrylamide gels, and then transferred onto a polyvinylidene fluoride (PVDF) membrane. After being pre-incubated in block solution at room temperature for 1 h, the PVDF membrane was incubated with rabbit anti-human NRP1 monoclonal antibody (1:1000) for 2 h at room temperature. After being washed 3 times with Tris-buffered saline (TBS) containing 0.5% Tween-20 (Solarbio, Beijing, China) for 10 min, the membrane was further incubated with goat anti-rabbit IgG secondary antibody (1:10,000) for 60 min at room temperature. Immunoreactive proteins were visualized using SuperSignal West Pico Chemiluminescent Substrate (Thermo Scientific, Waltham, MA, USA).

### 2.8. Confocal Fluorescence Imaging Study

For confocal fluorescence imaging, approximately 1 × 10^5^ cells (in 1 mL culture medium) were seeded in confocal dishes and cultured overnight. The nuclei were stained with Hoechst 33342 at the concentration of 1 mM for 10 min. Then the cells were incubated with Propidium Iodide (PI, 0.5 mM) or PAD (5 µΜ DOX) and LipoCy5 nanoparticles (0.1 µΜ Cy5) at 37 °C for 2 h, respectively. The cells were washed three times with PBS prior to observation. The confocal fluorescence imaging was performed on an Olympus FV1000-IX81 confocal-laser scanning microscope (Zeiss 710, Germany). For DOX, a FV5-LAMAR 488 nm laser was used for excitation, and the emission was collected between 520 and 620 nm. Cy5 was excited by a 640 nm laser and collected between 650 and 680 nm. Hoechst 33342 was excited by a FV5-LD405-2405 nm laser and collected at the range of 422 to 472 nm. All parameters of the microscope were set to be the same for comparisons of different cells with different treatments.

### 2.9. Cytotoxicity Analysis of LipoDOX Coadministered with PADiR

The cell viability assays were conducted by seeding PC-3 cells in 96-well plates at the density of 1 × 10^4^ cells/well and culturing for 24 h before experiments. When the cells had reached 70% confluence, they were incubated with DOX, LipoDOX, iRGD + LipoDOX, and PADiR + LipoDOX at different DOX concentrations for 20 h. The number of viable cells was determined using MTT assay. Briefly, the MTT solution was added to each well (100 µL/well) and incubated for 4 h. The solution was then carefully removed and 200 µL of dimethyl sulfoxide (DMSO) was added to each well. After 5 min of vibration mixing, the absorption at 570 nm was measured with an ELISA reader and the results were presented as mean ± SD % (*n* = 4) with the control group as 100% viability.

### 2.10. In Vitro Tumor Penetration Study

The in vitro tumor penetration efficacy of LipoDOX coadministered with PADiR peptosomes was studied using a multicellular tumor spheroid (MCTS) model. Briefly, 200 µL/well of cell suspension (1 × 10^4^ cells/mL) was transferred into the ultra-low attachment (ULA) 96-well round bottom plate. Then the plate was transferred into a cell incubator (37 °C, 5% CO_2_) and incubated for about 5 days to form the spheroids [28]. When the diameter of MCTS reached 500 μm, the culture medium was replaced with fresh medium supplemented with DOX, LipoDOX, and PADiR + LipoDOX at a DOX concentration of 10 μg/mL. The MCTS was further incubated for 5 h to allow the penetration of the nanoparticles into the spheroids. The penetration and distribution of DOX in the spheroids were examined using confocal fluorescence imaging.

### 2.11. In Vivo Distribution Analysis of Nanoparticles and Molecules Coadministered with APPA Self-Assembled Peptosomes

The fluorescent TRITC-containing APPA peptosomes (PAT) were prepared as the procedure mentioned above. All animal experiments were conducted in accordance with the institutional guidelines approved by the Institutional Ethical Committee of Animal Experimentation of National Center for Nanoscience and Technology. Animals received care in accordance with the Guidance Suggestions for the Care and Use of Laboratory Animals. The 5–7 week-old female BALB/c mice bearing 4T1 orthotopic tumors with the size of about 150 mm^3^ were divided into 10 groups with 3 mice in each group for in vivo fluorescence imaging. The mice in group No. 1 to No. 4 were injected with PAT (100 µL), PCDiR (100 µL), PAT (100 µL) + PCDiR (100 µL), and iRGD (4 µmol/kg) + PCDiR (100 µL), while the mice in group No. 5 to No. 10 were injected with LipoCy5 (100 µL), PAD (DOX: 0.1 µmol) + LipoCy5 (100 µL), iRGD (4 µmol/kg) + LipoCy5 (100 µL), FreeCy5 (100 µL), PAD (DOX: 0.1 µmol) + FreeCy5 (100 µL), and iRGD (4 µmol/kg) + FreeCy5 (100 µL) via tail vein with the Cy5 concentration at 2 μM, respectively. Before the injection, the two compounds were mixed together for the coadministration (Appendix A). After the injection, the fluorescence images of the mice for detecting DiR probes were acquired (excitation: 745 nm, emission: 800 nm) using an IVIS SPECTRUM in vivo imaging system at different time points. For detection of Cy5, the excitation wavelength was set at 640 nm and collected at 680 nm. After 8 h of injection, the mice were sacrificed and the main organs were harvested for ex vivo imaging.

### 2.12. PAD Peptosome Coadministration with Magnetosome and Fe_3_O_4_ for Magnetic Resonance Imaging (MRI)

The 5–7 week-old female BALB/c nude mice bearing PC-3 xenograft tumors with the size of about 150 mm^3^ were divided into 5 groups with 3 mice in each group for MRI imaging. The mice of each group were injected with magnotosome (Mag), PCD (DOX: 0.1 µmol) + Mag, iRGD (4 µmol/kg) + Mag, PAD (DOX: 0.1 µmol) + Mag, and PAD + Fe_3_O_4_ with the amount of Mag or Fe_3_O_4_ at 25 mg/kg. The two compounds were mixed together before the coadministration. The T_2_-weighted MRI imaging was conducted using a Bruker 7.0 T MR imaging system (Ettlingen, Germany) with the strength field of 7.0 T at 0 h, 2 h, 4 h, 5.5 h, and 7 h after the coadministration. Seven hours after the coadministration, the mice were sacrificed and the tumors were collected for Prussian blue staining analysis. To analyze the enrichment of small molecule, T_1_ contrast agent Gd-DTPA (0.15 mmol/kg) alone or together with PAD peptosome, were injected into three mice as the procedure described above, respectively. The T_1_-weighted MR imaging were conducted at 20 min, 40 min, and 2 h after the coadministration.

### 2.13. In Vivo Systemic Permeability Study

The 5–7 week old female BALB/c mice bearing 4T1 orthotopic tumors with the size of about 200 mm^3^ were divided into three groups with three mice in each group. The mice in each group were intravenously injected with 100 µL of PBS containing 1 mg of Evans blue (EB), iRGD (4 µmol/kg) + EB (1 mg), and PADiR (APPA:6 mg/kg) + trastuzumab (Tra, 1 mg), respectively. One hour after the injection, the mice were sacrificed and the main organs and tumors were collected. For Evans blue quantification, the dye was extracted from tissues in *N*,*N*-dimethylformamide for 24 h at 37 °C and the absorbance at 600 nm was quantified.

### 2.14. In Vivo Fluorescence Signal Colocalization of Padir Peptosome and the Coadministered Lipocy5 Liposome

The 5–7 week-old female BALB/c nude mice bearing HeLa xenograft tumors on their left flank and 4T1 xenograft tumors on the right flank with the tumor size of about 150 mm^3^ were divided into two groups with three mice in each group for fluorescence imaging. The three mice in one group were injected with anti-NRP1 antibody (50 μg) to block the function of NRP1 and those in the other group were injected with IgG (50 μg) as the control via the tail vein 15 min prior to the coadministration of PADiR and LipoCy5. Then the fluorescence signals of both Cy5 and DiR were acquired at 10 min, 2 h, 4 h, 6 h, and 9 h while the position of the mice stay fixed.

### 2.15. Immunofluorescence

The 5–7 week-old female BALB/c mice bearing 4T1 orthotopic tumors with the size of about 150 mm^3^ were divided into four groups with three mice in each group. The mice in each group were intravenously administered with LipoCy5 (100 µL), PAT (100 µL), PAT (100 µL) + free Cy5 (2 μM, 100 µL), and PAT (100 µL) + LipoCy5 (Cy5: 2 μM, 100 µL), respectively. Six hours after the injection, the tumors were excised and slices were prepared as the procedure described in our previous study [24]. The cell membranes were stained with FITC labeled anti-Na^+^/K^+^ ATPase antibody and the nuclei with DAPI. The slices were observed with a confocal fluorescence microscope.

### 2.16. Stimulated emission depletion (STED) Fluorescece Imaging to Visualize CoE Process

To investigate whether the coadministered nanoparticles were co-endocytosed into cells with APPA self-assembled peptosomes, stimulated emission depletion (STED) super-resolution fluorescence imaging microscopy was used to observe the cellular internalization process. To avoid the fast fluorescence quenching during the experiment, DiD and DiI fluorescent probes were used to construct the fluorescent nanoprobes PADiD (DiD containing APPA self-assembled peptosomes) and PCDiI (DiI containing APPC self-assembled peptosomes). Then the nanoprobes were incubated with cells in confocal dishes for 0.5 h, 1 h, and 2 h. After being washed 3 times, the cells were subject to STED imaging with DiI excitation at 550 nm and emission at5 65 nm, and DiD excitation at 644 nm and emission at 665 nm.

### 2.17. 4T1-H Orthotopic Tumor Therapy using Trastuzumab Coadministration with Peptosome

To study the therapeutic efficacy of trastuzumab coadministered with APPA self-assembled peptosome on mice bearing 4T1-H orthotopic tumors, the 4T1 cell line was transfected with pMH3-HER2 vector to construct the HER2 high expressing cell line (4T1-H). The expression of HER2 in the transfected cell line was analyzed by immunofluorescence staining as the procedure mentioned above. Then the 4T1-H cells were used to create the orthotopic tumor mouse model for the therapeutic study. The 5−7 week-old female BALB/c mice bearing 4T1-H orthotopic tumors with the size of about 60 mm^3^ were divided into four groups with five mice in each. The mice in each group were intravenously injected with PBS, trastuzumab (Tra: 6 mg/kg), iRGD (4µmol/kg) + Tra (6 mg/kg), and PADiR (APPA:6 mg/kg) + Tra (6 mg/kg) every other day, respectively. Fluorescence images were taken to show the distribution of Tra throughout the bodies using an IVIS SPECTRUM in vivo imaging system. The body weights of each mouse were measured before and after each injection. The tumor volumes were measured with a digital caliper and estimated by the formula (L × W^2^)/2, where L is the longest and W is the shortest diameter of the tumor. For further studies and humane reasons, the mice were sacrificed when the tumor volume in PBS group reached 800 mm^3^ after 6 injections. The livers, lungs, and kidneys were harvested for hematoxylin-eosin (H&E) staining. The distribution of Tra (Cy5 labeled) in the tumor tissue was analyzed by immunofluorescence imaging. The H&E staining slices were observed with a light microscope (EVOS, Life Technologies).

### 2.18. PC-3 Xenograft Tumor Treatment using LipoDOX Coadministered with Peptosome

The 5−7 week-old female BALB/c nude mice bearing PC-3 xenograft tumors with the size of about 100 mm^3^ were divided into four groups with five mice in each group. The mice in each group were intravenously injected with PBS, LipoDOX, iRGD (4 µmol/kg) + LipoDOX, and PADiR (APPA:6 mg/kg) + LipoDOX with the DOX dosage of 5 mg/kg every 3 days, respectively. The tumor volumes were measured and calculated as the procedure mentioned above. For further studies and humane reasons, the mice were sacrificed when the tumor volume in PBS group reached 700 mm^3^ after 5 coadministrations. The hearts, livers, lungs, and kidneys were harvested and subjected to hematoxylin-eosin (H&E) staining. Apoptosis of tumor cells was evaluated by H&E staining and terminal deoxynucleotidyl transferase dUTP nick end labeling (TUNEL). The TUNEL-positive cells were quantified by observing five random sections of tumor slices from each group by confocal microscopy. The data were presented as mean ± SD (*n* = 5).

## 3. Results and Discussion

### 3.1. Preparation and Characterization of Peptosomes, Magnetosomes, and Liposomes

APPA and APPC self-assembled peptosomes containing fluorescence probes were prepared by using the same method as described in our previous work [24] and the procedure described in the methods. The transmission electron microscopy (TEM) images of DiR- (DilC18(7) dialkyl carbocyanine membrane label) [29,30] containing APPA or APPC peptosome (PADiR or PCDiR), the DOX-containing APPA peptosome (PAD), and the tetramethylrhodamine isothiocyanate (TRITC)-containing APPA self-assembled peptosome (PAT) were shown in Appendix A. Together with the results of dynamic light scattering (DLS), all three types of peptosomes were shown to be negatively charged, had an average size of around 30 nm, and their size distributions were similar (Appendix A).

DiR is weakly fluorescent in water but highly fluorescent and quite photostable when incorporated into membranes. Therefore, its fluorescence signal can be used to reflect the integrity of peptosomes and estimate the in vivo plasma stability. The blood clearance of peptosome PADiR and peptide iRGD was compared by injecting the 5−7 week old female BALB/c nude mice with free Cy5, PADiR, and Cy5 labeled iRGD via the tail vein. Blood was then collected at different time points after the injection as described in the methods, and the fluorescence signals were detected using an IVIS Spectrum in vivo imaging system. The results suggest that the in vivo plasma half-life of the APPA self-assembled peptosome is approximately two hours, which is much longer than that of iRGD peptide (Appendix A) reported to be only eight min [31].

Magnetosomes, which are biomineralized by the magnetotactic bacteria, are reported to have high biocompatibility and minimal cytotoxicity [32,33]. Since magnetosomes have a narrow size distribution (30–50 nm) and a perfect crystal structure, using them as a T_2_-weighted magnetic resonance imaging (MRI) contrast agent offers greater advantages than using synthesized Fe_3_O_4_ nanoparticles (NPs) [25,34]. Magnetosomes were prepared according to the protocols provided in the methods. Fe_3_O_4_ NPs were purchased from Nanoeast Biotech (Nanjing, China). The TEM images of magnetosomes and the synthesized Fe_3_O_4_ NPs were shown in Appendix A. DLS (Zetasizer Nano ZS90, Malvern) was used to check the size and surface zeta potential of the magnetosomes and synthesized Fe_3_O_4_ NPs. The results shown in Appendix A indicated both materials were ready for further studies.

The Cy5- and DOX-containing liposomes, namely LipoCy5 and LipoDOX, were prepared based on a modified single-step preparation method [26], and the TEM images are shown in Appendix A. The size distribution and surface zeta potential of LipoCy5 and LipoDOX NPs were analyzed using DLS. The results in Appendix A indicate that LipoCy5 and LipoDOX NPs had a similar size distribution of 60–250 nm, and their surface zeta potentials were −24.9 mV and −21.6 mV, respectively.

All these data demonstrated that the peptosomes, magnetosomes, and liposomes were successfully prepared to be used in the following studies. In addition, we showed that the APPA self-assembled peptosomes were rather stable in the blood circulation.

### 3.2. Cytotoxicity of LipoDOX Coadministered with PADiR Peptosomes

The cytotoxicity of LipoDOX coadministered with PADiR peptosomes to PC-3 cell line with high NRP1 expression (Appendix A) was analyzed using MTT assay, as described in the methods. Briefly, cells were incubated with DOX, LipoDOX, iRGD + LipoDOX, and PADiR + LipoDOX at different DOX concentrations for 20 h. Then, the cell viability was estimated by MTT assay (Appendix A). The results indicated that the cytotoxicity to PC-3 cells for LipoDOX coadministered with both PADiR and iRGD were similar to that of free DOX and higher than LipoDOX alone. This suggested that with the coadministration of PADiR peptosomes and iRGD peptide, the cellular internalization of LipoDOX was significantly increased.

### 3.3. In Vitro Tumor Penetration of LipoDOX Coadministered with PADiR Peptosomes

The multicellular tumor spheroids (MCTS) of BT474 with high NRP1 expression were used to study the in vitro tumor penetration efficacy of LipoDOX coadministered with PADiR peptosomes. Briefly, MCTSs with the diameter of 500 μm were incubated with DOX, LipoDOX, and PADiR + LipoDOX NPs at the DOX concentration of 10 μg/mL at 37 °C for 6 h. The confocal fluorescence images (Appendix A) indicated that the LipoDOX coadministered with PADiR had a better tumor penetration efficacy than LipoDOX alone.

### 3.4. In Vivo Delivery of Nanoparticles Coadministered with APPA Self-Assembled Peptosomes to Tumors

To investigate the in vivo distribution of nanoparticles coadministered with APPA self-assembled peptosomes, the fluorescence imaging and MRI imaging studies were conducted using several different NPs labeled with appropriate probes.

Firstly, in vivo fluorescence imaging was performed to study whether PAT (TRITC-containing APPA peptosome) can improve tumor delivery of the coadministered PCDiR (DiR-containing APPC peptosome) which have no active-targeting property. The 5−7 week old female BALB/c mice bearing 4T1 orthotopic tumors at the size of approximately 150 mm^3^ were divided into four groups with three mice in each group. Mice of each group were intravenously injected with PAT, PCDiR, PAT + PCDiR, and iRGD+PCDiR, respectively. The DiR fluorescence images of the mice were acquired (Figure 1A, excitation: 745 nm; emission: 800 nm) using in vivo imaging system at 0 h, 2 h, 4 h, 6 h, and 8 h after the injection, as described in the methods. Mice were sacrificed 8 h after the injection, and the tumors and main organs were harvested for ex vivo imaging (Figure 1D). Figure 1A showed that the mice injected with PAT alone had no fluorescence signals because TRITC excites at 550 nm and emits at 570 nm, which is different from the wavelength set to detect DiR. In addition, the fluorescence signals of tumors in the PCDiR group were rather weak, even six h after the injection. Ex vivo fluorescence images indicated the signals of PCDiR were mostly distributed in the liver and spleen (Figure 1D). However, when the PCDiR was coadministered with PAT, the signals in tumors gradually increased and reached the highest intensity at six h after the injection (Figure 1A). By contrast, the fluorescence intensities of the liver and spleen were much lower than that of tumor. For mice coadministered with PCDiR and iRGD peptide, the tumor fluorescence signals were approximately three times less than that of PCDiR + PAT group (Figure 1G).

Secondly, the tumor delivery efficacy was compared between liposome-based nanoprobe LipoCy5 and free Cy5 when both were coadministered with PAD (DOX-containing APPA self-assembled peptosome). The LipoCy5 nanoprobe was prepared according to the procedure described in the methods. The 5−7 week old female BALB/c mice bearing 4T1 orthotopic tumors with the size of about 150 mm^3^ were divided into six groups, with three mice in each group, for in vivo fluorescence imaging. The mice in each group were injected with LipoCy5, PAD + LipoCy5, iRGD + LipoCy5, FreeCy5, PAD + FreeCy5, and iRGD + FreeCy5 via the tail vein, respectively. After the injection, fluorescence images of the mice were acquired to detect Cy5 (excitation: 640 nm; emission: 680 nm) at different time points (Figure 1B,C). Mice were then sacrificed eight h after the injection, and the tumors and main organs were harvested for ex vivo fluorescence imaging (Figure 1E,F). For the mice injected with LipoCy5 nanoprobes alone, only a very weak fluorescence signal was detected in tumors. However, the nanoprobes were significantly enriched in tumors when coadministered with PAD, and the fluorescence signal of the tumor was about 10 times higher than that of LipoCy5 alone (Figure 1B,E,H). For the mice injected with FreeCy5, PAD + FreeCy5, and iRGD + FreeCy5, the fluorescence signals were strong throughout the bodies, with the liver exhibiting the highest signal shortly after the injection. Then the signal gradually decreased, and only very weak fluorescence signal could be detected in kidneys six h after the injection (Figure 1C,F,I). This result indicated the delivery of small molecules to tumors was not significantly improved by the coadministration with PAD peptosomes or iRGD peptides, which is consistent with the replication study by Mantis et al. but conflicts with the others [22,23,35,36,37]. Differences in the cell line, the injected reagents, and the coadministration procedures between our work and others can contribute to the inconsistency of the outcome. Further investigations and comparisons under similar conditions by more research groups are necessary. To confirm the inefficiency of peptosomes to improve the delivery of small molecules, we performed T_1_-weighted MRI imaging of tumor-bearing mice treated with gadopentetic acid (Gd-DTPA) alone and those with Gd-DTPA and PAD coadministration (Appendix A). No significant differences of the images indicated PAD coadministration did not improve the tumor uptake of Gd-DTPA. One reason might be that small molecules are more easily metabolized and degraded compared to nanoparticles [38,39]. We will explain the other reason influencing the effect of peptosomes on different coadministered compounds in the following study.

Thirdly, we extended the study of the tumor delivery efficiency influenced by APPA peptosome to inorganic nanoparticles: magnetosome (Mag) and Fe_3_O_4_ NPs. The 5−7 week old female BALB/c nude mice bearing PC-3 xenograft tumors with the size of about 150 mm^3^ were divided into five groups with three in each one. Mice in each group were injected with Mag, PCD + Mag, iRGD + Mag, PAD + Mag, and PAD + Fe_3_O_4_ with the Mag or Fe_3_O_4_ at the dosage of 25 mg/kg. The MRI imaging was conducted on a Bruker 7.0 T MRI system (Ettlingen, Germany) with the strength field of 7.0 T at 0 h, 2 h, 4 h, 5.5 h, and 7 h after the injection. Seven hours after the injection, the mice were sacrificed, and the tumors were collected for Prussian blue staining analysis. The results in Figure 2A show that the tumors of mice injected with Mag alone or Mag + PCD have no significant color decay after the administration. The tumors of mice injected with Mag + iRGD show rather mild color decay 7 h after the injection. However, significant color decay was observed for magnetosomes and Fe_3_O_4_ NPs coadministered with PAD peptosomes 7 h after the injection (Figure 2A). The normalized signal-to- noise ratio (SNR) of xenograft tumors 7 h after the coadministration with PAD peptosome was about 50% of that before the administration (Figure 2C). The Prussian blue staining of tumor slices further verified that both the magnetosome and Fe_3_O_4_ NPs coadministered with PAD peptosome were enriched in xenograft tumors (Figure 2B).

The above results from both in vitro and in vivo studies indicate that the coadministration with APPA self-assembled peptosomes enhances the delivery of nanoparticles, but not non-targeting small molecules, to integrin αvβ3 and NRP1-positive tumors. In the following studies, we set out to investigate the mechanisms that mediate the tumor specific enrichment of nanoparticles when coadministered with APPA self-assembled peptosomes.

### 3.5. Mechanisms of Tumor Specific Enrichment of Nanoparticles with APPA Peptosome Coadministration

To determine whether the enhanced tumor enrichment of nanoparticles was due to impaired cell membranes caused by peptosomes, propidium iodide (PI) staining was performed to analyze the membrane integrity. HUVEC and 4TI cells with high expression of both integrin αvβ3 and NRP1 receptors (Appendix A) were incubated with PADiR peptosomes and PI for different time lengths prior to the imaging [27,40]. Appendix A show that the nuclei were not stained by PI even after 3 h of incubation, suggesting that the cell membranes were intact. The enhanced tumor enrichment of coadministered nanoparticles was not caused by impaired cell membranes. We then investigated the vascular permeability by measuring tumor-specific enrichment of Evans blue coadministered with PADiR peptosomes. The result indicated that both PADiR peptosomes and iRGD peptide induced vascular leakage (Appendix A).

It is reported that CendR peptides enter the cells through an endocytic pathway resembling macropinocytosis, a process that is mediated by NRP1 receptor [14,27,41,42]. Binding of APPA self-assembled peptosomes to NRP1 highly expressed tumor cells may activate the NRP1-mediated endocytosis pathway [35,38] triggering cellular membrane to actively undergo invagination. While the coadministered nanoparticles are attached or close to the membrane, they may be wrapped and entered the cells simultaneously with APPA self-assembled peptosomes. We call this process “co-endocytosis (CoE)”, which leads to faster internalization of compounds bound to the cell membrane or accumulated around the invagination site. To verify this hypothesis, in vitro and in vivo colocalization studies were carried out, including using stimulated emission depletion (STED) microscopy to observe the CoE process in super resolution. 

Firstly, the colocalization of PAD peptosomes and LipoCy5 nanoprobes was studied using in vitro fluorescence imaging. Briefly, approximately 1 × 10^5^ cells were seeded in confocal dishes and cultured overnight. The cells were incubated with PAD peptosomes and LipoCy5 nanoprobes simultaneously at 37 °C for 2 h. The results in Figure 3B indicate that the fluorescence intensities of both PAD and LipoCy5 were high and colocalize well in both 4T1 and HUVEC cells, especially on the cell membrane. However, only weak fluorescence signal of both PAD and LipoCy5 were detected in HeLa cells due to low NRP1 expression. In addition, when NRP1 receptors of HUVEC cells were knockdown using siRNA, the fluorescence signals were much lower. This means that PAD and LipoCy5 nanoparticles enter the cells through the NRP1 mediated pathway.

Secondly, in vivo colocalization of PADiR and the coadministered LipoCy5 NPs were analyzed. The 5−7 week old female BALB/c nude mice bearing HeLa xenograft tumors (negative control) on their left flanks and 4T1 xenograft tumors on their right flanks with tumor sizes of about 150 mm^3^ were divided into two groups, with three mice in each one. The mice in one group were injected with anti-NRP1 antibody (50 μg) to block the activity of NRP1, and the mice in the other group were injected with IgG (50 μg) as a control via the tail vein 15 min prior to the coadministration of PADiR and LipoCy5 NPs. The representative images acquired for Cy5 and DiR are shown in Figure 3A, and the results indicate that both PADiR and LipoCy5 NPs were enriched in 4T1 but not in HeLa xenograft tumors. The fluorescence signal of PADiR and LipoCy5 were colocalized at 4T1 tumor sites. Pre-injection of anti-NRP1 antibody but not IgG, blocked the enrichment of both PADiR and LipoCy5 NPs in 4T1 xenograft tumors. The results confirm that the improved tumor delivery of both PADiR and LipoCy5 NPs are mediated by NRP1 receptor.

We further used immunofluorescence staining to check the colocalization of APPA peptosomes and the coadministered LipoCy5 NPs in tumor tissues from the 5−7 week old female BALB/c mice bearing 4T1 orthotopic tumors. The cell membranes were stained with FITC labeled anti-Na^+^/K^+^ ATPase antibodies. Slices were observed with a confocal microscope. Figure 3C shows that treatment of LipoCy5 NPs alone or free Cy5 coadministered with PAT resulted in no Cy5 signals in tumors. However, when the LipoCy5 was coadministered with PAT, Cy5 intensities were significantly high in tumor tissues. And the signals of LipoCy5 overlapped with that of peptosomes as well as the signals of FITC on the cell membrane.

Finally, we used STED super-resolution fluorescence imaging microscopy to visualize the CoE process. Before this, we verified the improved endocytosis of PCDiI (DiI-containing APPC self-assembled peptosomes) when coadministered with PADiD (DiD-containing APPA peptosomes). Incubation of PCDiI alone with PC-3 cells resulted in very weak binding of the peptosomes, even after 1.5 h of incubation (Appendix A). When PCDiI was coadministered with PADiD, both signals became strong and well colocalized on the cell membrane (Figure 3D), with the signals of PADiD (red) a bit higher in the intracellular front edge at 0.5 h and 1 h. This indicates that both of them attached to the cell membranes and then gradually entered the cells after longer time incubation, despite that some PADiD seem to enter the cells a bit earlier. Most of PADiD and PCDiI fluorescence signals were well-colocalized, especially on the cell membrane, which means they were close to each other during the cell-entering process. After 2 h of incubation, more PADiD and PCDiI peptosomes were observed inside the cells. These results are consistent with former colocalization studies and explain the enhanced tumor enrichment of nanoparticles coadministered with APPA self-assembled peptosomes. Activation of NRP1 pathway by APPA peptosomes promotes endocytosis, resulting in compounds that bind to the cellular membrane or gather around the membrane invagination site to be wrapped in vesicles simultaneously, and thus enter the cells more efficiently. Nanoparticles have the passive-targeting ability due to the enhanced permeability and retention (EPR) effect and are known to bind to scavenger receptors on the cellular membrane [43]. However, non-targeting small molecules do not have receptors on or gather around the membrane to be wrapped in vesicles during the membrane invagination of the CoE process. Accordingly, the delivery of antibodies that target tumors actively by binding to tumor cell membranes can be also improved by peptosome coadministration as we will show in the following study (Figure 4).

### 3.6. 4T1-H Orthotopic Tumor Therapy using Coadministration Strategy

We further tested the therapeutic effect of antitumor compounds with APPA peptosome coadministration. Trastuzumab (Tra), which works by binding to the HER2 receptor, is a monoclonal antibody used to treat breast cancer. At first, 4T1 cells were transfected with pMH3-HER2 vector to construct the HER2 high-expressed cell line (4T1-H). Then the 4T1-H cells were orthotopically inoculated to 5−7 week old female BALB/c mice. The expression of HER2 receptor for 4T1-H orthotopic tumors was confirmed by immunofluorescence staining (Appendix A). The mice bearing 4T1-H tumors with the size of about 60 mm^3^ were divided into four groups, with five mice in each group. The mice in each group were intravenously injected with PBS, Tra, iRGD + Tra, and PADiR + Tra with the dosage of Tra at 6 mg/kg every other day. After six injections, the mice were sacrificed, and the livers, lungs, and kidneys were harvested for hematoxylin-eosin (H&E) staining (Figure 4A). The body weights of all mice remained close to their starting status during the therapeutic study (Figure 4B). Photos were taken to show the growth of the tumors during the treatments (Figure 4D). Tumor volumes over time were further calculated for each group as described in the methods. The results demonstrated Tra coadministered with PADiR significantly inhibits the growth of tumors when compared to PBS, Tra and iRGD + Tra (Figure 4C). Metastatic tumors were observed in the liver and spleen 21 days after the inoculation for the mice in PBS group (Appendix A). After the mice were sacrificed, the excised tumors (Figure 4F) were weighted and the average values were shown in Figure 4E. The tumor sizes of mice treated with PADiR + Tra were 4 times smaller than those of PBS group and 2–3 times smaller than those treated with Tra and iRGD + Tra.

Substantial cell morphology changes were observed for the tumor slices from the mice treated with PADiR + Tra after H&E staining. Most of the tumor cells were shrunken and loosely connected with each other compared to those from the mice treated with PBS, Tra, and iRGD + Tra (Figure 4G). Immunofluorescence imaging of tumor slices showed that Tra antibodies coadministered with PADiR were significantly enriched in tumor tissue compared to those treated with Tra alone or iRGD + Tra (Figure 4H). The H&E staining results showed that the cell morphology of the lungs, livers, and kidneys from the mice of all four groups exhibited no obvious changes after the treatment (Appendix A). 

### 3.7. Anti-Tumor Efficacy of LipoDOX Coadministered with PADiR Peptosomes

The anti-tumor efficacy of LipoDOX coadministered with PADiR was studied on 5−7 week old female BALB/c nude mice bearing PC-3 xenograft tumors at the size of about 100 mm^3^. Briefly, the mice were divided into four groups, with five mice in each group. The mice in each group were intravenously injected with PBS, LipoDOX, iRGD + LipoDOX, and PADiR + LipoDOX every three days, respectively (Figure 5A). After five injections, the mice were sacrificed, and the hearts, livers, lungs, kidneys, and tumors were harvested for histological analysis. As shown in Figure 5B, the body weights of all mice remained close to their starting status during the therapeutic study. Terminal deoxynucleotidyl transferase dUTP nick end labeling (TUNEL) staining was performed to determine DNA fragmentation by labeling the 3′-hydroxyl termini in the double-stranded DNA breaks that were generated during apoptosis [44]. Coadministration of LipoDOX with PADiR peptosomes resulted in significant inhibition of tumor growth compared to administration of PBS or LipoDOX alone. LipoDOX coadministered with iRGD peptide also inhibited the growth of tumors, but the efficacy is lower than with PADiR coadministration (Figure 5C–F).

H&E staining of tumor slices from the mice treated with PADiR + LipoDOX showed substantial cell morphology changes. Most of the cells were shrunken and loosely connected with each other (Figure 5G) compared to those in the other three groups. TUNEL staining analysis indicated that most tumor cells from the mice treated with PADiR + LipoCy5 were TUNEL positive, suggesting the cells were apoptotic (Figure 5G). By contrast, tumor slices from the mice treated with PBS were TUNEL negative. The percentage of TUNEL-positive tumor cells in the PADiR + LipoDOX group was 65.1 ± 5.8%, which is much higher than those treated with LipoDOX alone (7.4 ± 3.1%) (Figure 5H). The results of H&E staining showed that the cell morphology of hearts, lungs, livers, and kidneys taken from the mice in all four groups exhibited no obvious changes after the treatment (Appendix A). All these data demonstrated the improved tumor delivery and therapeutic efficacy of nanodrugs coadministered with APPA self-assembled peptosomes.

## 4. Conclusions

In this study, we investigated the tumor delivery efficiency of nanoparticles and antibodies coadministered with APPA self-assembled peptosomes. The enrichment of nanoparticles and antibodies in tumors is largely improved through the NRP1-mediated CoE pathway, which is activated by APPA self-assembled peptosomes. The binding of peptosomes to NRP1 receptors triggers the cellular membrane invagination, which leads to active internalization of the coadministered antibodies or nanoparticles bound to cell membranes. However, the cellular uptake of non-targeting small molecules is not improved as these compounds do not bind to or gather close to the cell membrane for the CoE process to happen.

The effectiveness of APPA self-assembled peptosomes as nanoparticle adjuvant may be due to several of their characteristics: (1) the plasma half-life of the peptosome is much longer than that of NRP1-binding peptides, such as iRGD; (2) the EPR effect of peptosome promoted the enrichment in tumor vessels; (3) each peptosome consists of multiple ligands for binding with NRP1 and integrin αvβ3 receptors to ensure the activation of NRP1-mediated endocytosis pathway. Nevertheless, more studies can be performed to optimize the method, such as the interval between administrations and the ratios of the amount of nanoparticles and peptosomes. Overall, the peptosome coadministration strategy has great potential to be applied in improving tumor-specific cargo delivery. Together with its low cytotoxicity, high biocompatibility, and low-cost, the APPA self-assembled peptosomes could be developed for clinical applications.

## Figures and Tables

**Figure 1 biomolecules-09-00172-f001:**
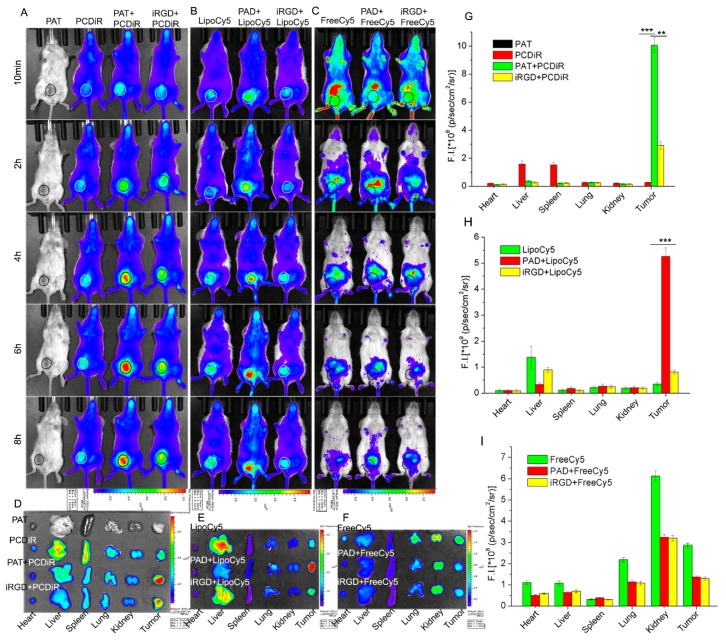
Enhanced tumor enrichment of nanoparticles coadministered with APPA self-assembled peptosomes. In vivo fluorescence imaging of mice to compare tumor delivery efficiency of APPC self-assembled nanoprobes coadministered with PAT (**A**), Cy5-containing liposomes coadministered with PAD (**B**), and free Cy5 fluorescent probes coadministered with PAD (**C**). Tumors in (A), (B) and (C) panels are highlighted with circles. Ex vivo fluorescence imaging of tumors and main organs harvested (**D**), (**E**) and (**F**) and their quantified intensities (**G**), (**H**) and (**I**) from the mice shown in (A), (B) and (C), respectively. Results are shown as mean fluorescence intensity ±SD (*n* = 3), ** *p* < 0.01, *** *p* < 0.001.

**Figure 2 biomolecules-09-00172-f002:**
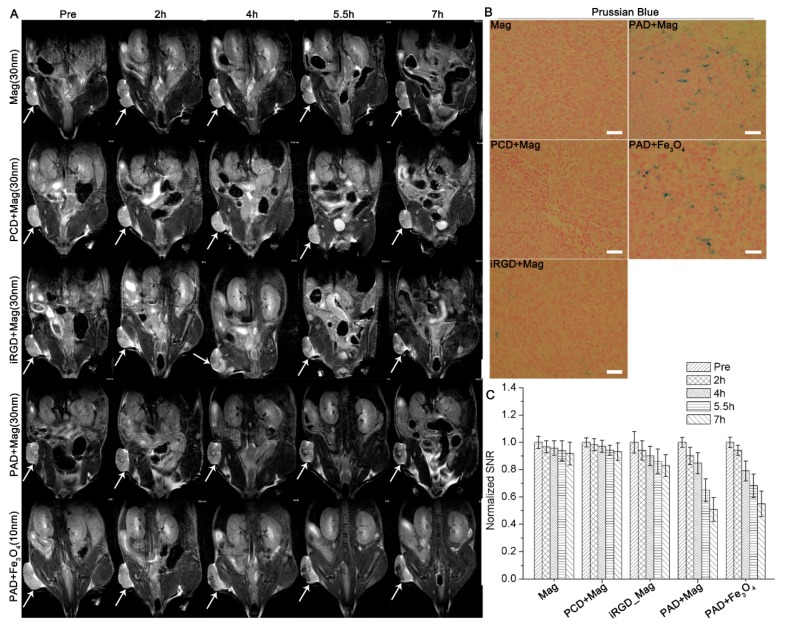
T_2_-weighted MRI imaging of BALB/c nude mice bearing PC-3 xenograft tumors before and after the injections. (**A**) MRI images of the mice before and after injected with Mag, PCD + Mag, iRGD + Mag, PAD + Mag, and PAD + Fe_3_O_4_ (10 nm) at 2 h, 4 h, 5.5 h, and 7 h. The white arrows indicate xenograft tumors; (**B**) Prussian blue staining of tumor slices from the mice 7 h after the injection (scale bars: 25 µm); (**C**) the normalized signal-to-noise ratio (SNR) ratios of T_2_ signal for the tumors in each group at different time points.

**Figure 3 biomolecules-09-00172-f003:**
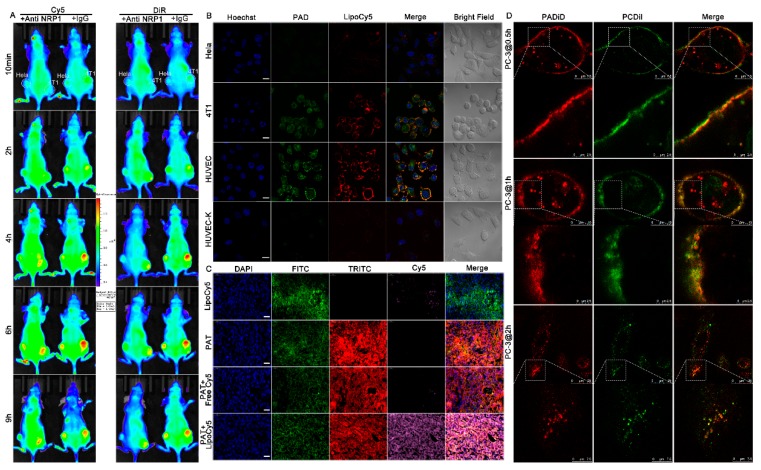
NRP1-mediated CoE process for enhanced tumor enrichment of nanoparticles coadministered with APPA self-assembled peptosomes. (**A**) In vivo colocalization analysis of coadministered PADiR and LipoCy5 nanoparticles. HeLa xenograft tumors are on the left flank of the mice and 4T1 xenograft tumors are on the right. The fluorescence images of Cy5 (left) and DiR (right) probes were taken for each mouse treated with either anti-NRP1 antibody or IgG; (**B**) confocal fluorescence images of HeLa, 4T1, HUVEC, and HUVEC-K (siRNA-mediated NRP1 knockdown of HUVEC) cells incubated with PAD and LipoCy5 nanoparticles (Hoechst: blue, PAD: green, LipoCy5: red, scale bars: 25 µm); (**C**) colocalization analysis of PAT and LipoCy5 in tumor tissues using immunofluorescence staining (DAPI: blue, FITC-anti-Na^+^/K^+^ ATPase antibodies: green, TRITC: red, Cy5: pink, and scale bars: 20 µm); (**D**) stimulated emission depletion (STED) fluorescence images showing the details of the CoE process in super resolution (PADiD: red and PCDiI: green).

**Figure 4 biomolecules-09-00172-f004:**
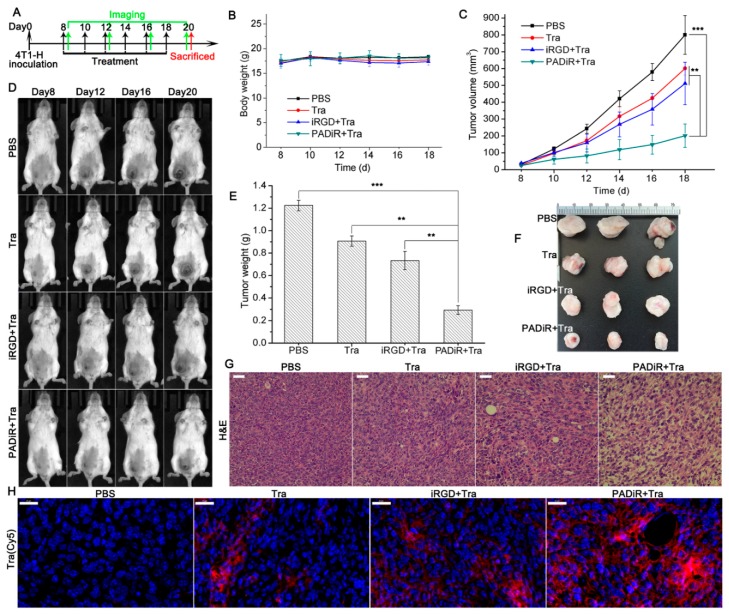
Enhanced antitumor effects of trastuzumab (Tra) coadministered with PADiR peptosomes. (**A**) Scheme of coadministration therapy: 4T1-H orthotopic tumor-bearing mice were injected with Tra and PADiR every other day; (**B**) the body weights of the mice in each group during the therapy. Data are shown as mean ±SD (*n* = 5); (**C**) the growth curves of tumors from the mice treated with phosphate-buffered saline (PBS), Tra, iRGD + Tra, and PADiR + Tra in the therapy study. Data are shown as mean ±SD (*n* = 5), ** *p* < 0.01, *** *p* < 0.001; (**D**) representative images showing tumor growth during the therapy; (**E**) average tumor weights from the mice in each group after the therapy. Data are shown as mean ±SD (*n* = 5), ** *p* < 0.01, *** *p* < 0.001; (F) representative images of excised tumors from the mice treated with PBS, Tra, iRGD + Tra, and PADiR + Tra; (**G**) histological analysis using hematoxylin-eosin (H&E) staining of tumor slices from each group after the treatment; scale bars: 50 µm; (**H**) analysis of Tra (Cy5 labeled) enrichment in tumor tissues using immunofluorescence staining (DAPI: blue, Tra: red, and scale bars: 50 µm).

**Figure 5 biomolecules-09-00172-f005:**
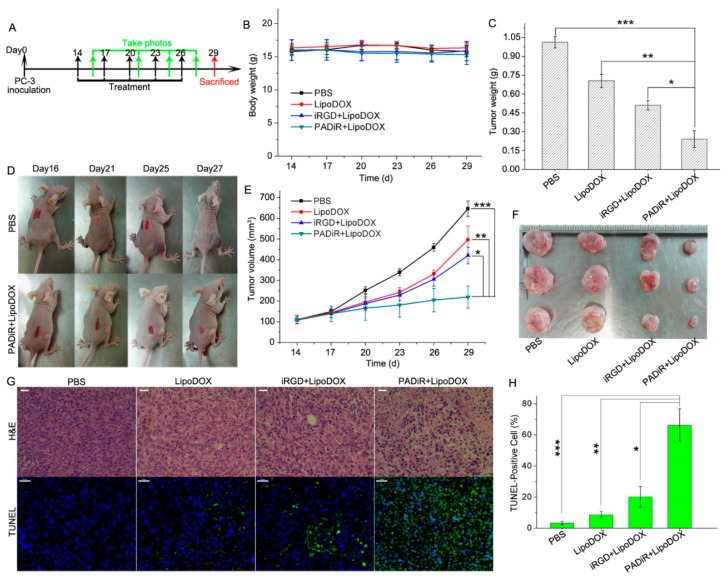
Improved antitumor effects of DOX-loaded liposomes coadministered with PADiR peptosomes. (**A**) Scheme of PC-3 xenograft tumor therapy with peptosome coadministration; (**B**) body weights of the mice in each group during the therapy study. Data are shown as mean ±SD (*n* = 5); (**C**) average tumor weights of the mice for each group after the therapy. Data are shown as mean ±SD (*n* = 5), * *p* < 0.05, ** *p* < 0.01, *** *p* < 0.001; (**D**) representative images of the mice treated with PBS and PADiR + LipoDOX showing the overtime growth of tumors; (**E**) growth curves of the tumors from mice treated with PBS, LipoDOX, iRGD + LipoDOX, and PADiR + LipoDOX. Data are shown as mean ±SD (*n* = 5), * *p* < 0.05, ** *p* < 0.01, *** *p* < 0.001; (**F**) representative images of excised tumors of the mice for each group; (**G**) H&E and transferase dUTP nick end labeling (TUNEL) staining of tumor slices from each group after the therapy study (DAPI: blue, TUNEL: green, and scale bars: 50 µm); (**H**) quantification of TUNEL-positive signals in the tumors cells of each group. Data are shown as mean ±SD (*n* = 5); * *p* < 0.05, ** *p* < 0.01, *** *p* < 0.001.

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
