# Peer review of "Peptosome Coadministration Improves Nanoparticle Delivery to Tumors through NRP1-Mediated Co-Endocytosis"

_biomolecules, 2019, doi:10.3390/biom9050172_

Round 1
Reviewer 1 Report
This work is very interesting. This systematic study has demonstrated that co-administration of targeting peptosomes and nanoparticles significantly improves the delivery of nanoparticles into tumour sites with enhanced anti-tumour performance. The mechanisms are also studied and neuropilin-1 medicated co-endocytosis is proposed to explain the observed co-administration effect. The conclusions are well-supported by the presented data. Thus, I would recommend acceptance of this work in Biomolecules after addressing the following concerns.
1) When the peptosomes and nanoparticles are mixed before injection (e.g. PAT+PCDiR, or PAD+LipoCy5), are there any changes in terms of particle morphology or structure?
2) For Figure 3A, how about the quantified data of the fluorescence signals in two tumours?
Author Response
Reviewer 1:
This work is very interesting. This systematic study has demonstrated that co-administration of targeting peptosomes and nanoparticles significantly improves the delivery of nanoparticles into tumour sites with enhanced anti-tumour performance. The mechanisms are also studied and neuropilin-1 medicated co-endocytosis is proposed to explain the observed co-administration effect. The conclusions are well-supported by the presented data. Thus, I would recommend acceptance of this work in Biomolecules after addressing the following concerns.
1) When the peptosomes and nanoparticles are mixed before injection (e.g. PAT+PCDiR, or PAD+LipoCy5), are there any changes in terms of particle morphology or structure?
Answer:
Thanks for the reviewer’s comments. This is a good suggestion to look at the stabilities of peptosomes and nanoparticles when mixed together. The TEM images of PAT+PCDiR and PAD+LipoCy5 10min and 1.5h after the mixing were shown as following and in Figure S4. The results indicate that the mixing of PAT and PCDiR peptosomes caused minimal morphology changes of both, even at 1.5 h after the incubation. This may be attributed to the similar surface structure and zeta potential of PAT and PCDiR peptosomes. Similarly, for PAD and LipoCy5 nanoparticles, no significant morphology changes were observed 10min after the mixing. However, the outline of LipoCy5 became fuzzy and some PAD peptosomes seem to attach to the surface of LipoCy5 nanoparticles 1.5 h after the mixing. This may be due to the interactions between phosphor groups of LipoCy5 and amino acid side chains exposed on the surface of PAD. Nevertheless, the coadministration of peptosomes+nanoparticles was performed right after the two compounds were mixed. Once injected into the blood, the concentrations of both nanoparticles decrease and the interaction should be eliminated. In the meantime, the blood flow may also reduce the interaction.
2) For Figure 3A, how about the quantified data of the fluorescence signals in two tumours?
Answer:
Thank you. We are sorry that we didn’t take fluorescence images of the excised organs and tumors in this assay. The purpose of Figure 3A is to show that the fluorescence signal of APPA peptosomes and the coadministered nanoparticles can be colocalized at the tumor sites in vivo. However, we do quantify the fluorescence signals of APPA peptosomes with several coadministered nanoparticles in the tumors and main organs in Figure 1D-E.

Reviewer 2 Report
Manuscript entitled “Peptosome coadministration improves nanoparticle delivery to tumors through NRP1-mediated co-endocytosis” by Xiang et al. report the use of peptosome to improve delivery of nanoparticles as well as antibody in the cancer. This work is based on the author recent work (Adv. Therap. 2019, 2, 1800107) where they reported the synthesis and evaluation of peptosome for delivery of Dox and other probe using invitro and invivo cancer model. Here in this manuscript author successfully reported the use of same peptosome (APPA) to the delivery of nanoparticles and antibody. The self assembled peptosomes which form a size of 30 nm were used in the studies which matches with recent reported studies. Author provided numerous assay (invitro) to test the cytotoxicity of their delivery system using overexpressed cell lines (4T1, HUVEC, PC-3 and BT474) followed by invivo assay using 5−7 week-old female BALB/c mice. Very interesting cytotoxicity, plasma stability, confocal studies, invitro tumor penetration study were performed to show that peptosome with or with labeling agents are safe and penetrate in the cancer cells. Followed by invivo studies successfully demonstrated application of peptosome coadministration in the delivery of nanoparticle and antibody. More importantly, the mechanism of improved delivery of nanoparticle or antibody with peptosome were fully elucidated with supportive data as co-endocytosis. Also, the manuscript is well written and is found suitable for publication in the Biomolecules after revision with following comments.
Author should provide some structure or schematic of peptosome (APPA) so that reader can understand it carefully.
Why only one sex of mice (female, 5−7 week-old female BALB/c nude mice) were used in all the studies?
Line 88, How the weighing of micromilligram amount were performed? Provide specific details of instrument and specific handling of drugs.
Line 111, Provide amount/quantities of Cy5-NHS and iRGD peptide used in the reaction. How this peptide was purified.
Line 134, Provide strain number and sources of the cell lines used in the studies. Similarly sources of mice need to mentioned in the manuscript.
Line 174 Why DMSO was added to each well ?
Line 385, Either experimental or literature evidence should be the provided for the reason of “easily metabolized and degraded”.
Author Response
Reviewer 2:
Manuscript entitled “Peptosome coadministration improves nanoparticle delivery to tumors through NRP1-mediated co-endocytosis” by Xiang et al. report the use of peptosome to improve delivery of nanoparticles as well as antibody in the cancer. This work is based on the author recent work (Adv. Therap. 2019, 2, 1800107) where they reported the synthesis and evaluation of peptosome for delivery of Dox and other probe using invitro and invivo cancer model. Here in this manuscript author successfully reported the use of same peptosome (APPA) to the delivery of nanoparticles and antibody. The self assembled peptosomes which form a size of 30 nm were used in the studies which matches with recent reported studies. Author provided numerous assay (invitro) to test the cytotoxicity of their delivery system using overexpressed cell lines (4T1, HUVEC, PC-3 and BT474) followed by invivo assay using 5−7 week-old female BALB/c mice. Very interesting cytotoxicity, plasma stability, confocal studies, invitro tumor penetration study were performed to show that peptosome with or with labeling agents are safe and penetrate in the cancer cells. Followed by invivo studies successfully demonstrated application of peptosome coadministration in the delivery of nanoparticle and antibody. More importantly, the mechanism of improved delivery of nanoparticle or antibody with peptosome were fully elucidated with supportive data as co-endocytosis. Also, the manuscript is well written and is found suitable for publication in the Biomolecules after revision with following comments.
Author should provide some structure or schematic of peptosome (APPA) so that reader can understand it carefully.
Answer:
Thank you. This is a good concern. The structure or schematic of peptosome (APPA) was shown in Scheme 1 in our previous published paper (https://onlinelibrary.wiley.com/doi/10.1002/adtp.201800107) as shown in the following figure. We have described it in the revised manuscript on line 61 to help the readers better understand it.
Why only one sex of mice (female, 5−7 week-old female BALB/c nude mice) were used in all the studies?
Answer:
Thanks for the reviewer’s comments. For 4T1 (breast cancer) and Hela (cervical carcinoma) cell lines, they are much easier to construct tumor models on female mice. For PC-3 (prostate carcinoma) cells, both female and male mice are ok for xenograft model construction. Here we chose female mice because of the limitations of feeding conditions.
Line 88, How the weighing of micromilligram amount were performed? Provide specific details of instrument and specific handling of drugs.
Answer:
Thanks. To make it cleaer, we have revised the method (line 92-95) describing the making of self-assembled peptosomes. “1 mg of all three compounds that were to be encapsulated in peptosomes (DOX, DiR or TRITC) were weighed by Precision Electronic Balance (Sartorius-BT25S). Each compound was dissolved in 100uL of DMSO to make a stock solution at 10mg/mL. Then 1 mg of peptide was dissolved in the above stock solution in the amount of 5 μL of DOX, 2 μL of DiR and 10 μL of TRITC, respectively. DMSO was added to bring the solution to 10 uL. The mixture was then added to 1 mL PBS ….”
Line 111, Provide amount/quantities of Cy5-NHS and iRGD peptide used in the reaction. How this peptide was purified.
Answer:
Thank you. The reaction was conducted by mixing 0.3 mg of Cy5-NHS and 0.5 mg of iRGD (molar ratio 1:1) in 1 mL sodium bicarbonate buffer (pH 6.5). Then the Cy5 labeled peptide were purified by using a Hitachi HPLC system (L-7100, Japan) on a TSK gel ODS-100V column (150 mm×4.6 mm) at a flow rate of 2 mL min-1 with the gradient of 5-80% acetonitrile containing 0.1% Trifluoroacetic acid (TFA) from 0 to 25 min. We have added the detail description to the revised manuscript on line 120.
Line 134, Provide strain number and sources of the cell lines used in the studies. Similarly sources of mice need to mentioned in the manuscript.
Answer:
Thanks. The strain numbers of 4T1, HUVEC, PC-3, BT474 and Hela cell lines are ATCC CRL-2539, ATCC CRL-1730, 3111C0001CCC000115, 3142C0001000000314 and ATCC CCL-2, respectively. The 4T1, HUVEC and Hela cell lines were bought from ATCC. The PC-3 and BT474 cell lines were bought from NICLR (National Infrastructure of Cell Line Resource, China). The mice used in this study were bought from Charles River (Beijing, China). These have been added to the revised manuscript on line 86.
Line 174 Why DMSO was added to each well ?
Answer:
Thanks. We follow the protocol of MTT assay to measure the cell viability. The DMSO was added to dissolve the formazan produced by the mitochondria of living cells, so as to estimate the number of the living cells.
Line 385, Either experimental or literature evidence should be the provided for the reason of “easily metabolized and degraded”.
Answer:
Thanks for the good suggestion. Two references were provided in line 399 in the revised manuscript. At the same time, the in vivo fluorescence imaging results of FreeCy5 and LipoCy5 (Figure 1B, C), and the T1-weighted MRI imaging results using gadopentetic acid (Gd-DTPA) (Figure S7) indicated the short plasma half -life of small molecules (FreeCy5 and Gd-DTPA ).

Reviewer 3 Report
The manuscript reports on the synthesis of APPA and APPC self-assembled peptosomes, magnetosomes and liposomes and the study of their microstructural (TEM) and drug delivery properties. The objective is to use the co-administration of the particles with the peptosomes in order improve the delivery of the particles and antibodies to the integrin αvβ3 and neuropilin-1 tumors, with reduced side-effects (such as reduced impact on cell membranes). The cytotoxicity, drug delivery and tumor penetration were studied. A mechanism is developed to explain the observed results. A detailed study of the efficiency of tumor peptide co-delivery of the magnetic nanoparticles and of antibodies, as well as the characterization of their anti-tumor efficacy, were performed. The manuscript, with its detailed and extensive characterizations, has original results and can be published as is.
Author Response
Thanks for the positive comments !